# Non-Invasive Assessment of Arterial Stiffness: Pulse Wave Velocity, Pulse Wave Analysis and Carotid Cross-Sectional Distensibility: Comparison between Methods

**DOI:** 10.3390/jcm11082225

**Published:** 2022-04-15

**Authors:** Paolo Salvi, Filippo Valbusa, Anna Kearney-Schwartz, Carlos Labat, Andrea Grillo, Gianfranco Parati, Athanase Benetos

**Affiliations:** 1Cardiology Unit, Istituto Auxologico Italiano, IRCCS, 20100 Milan, Italy; gianfranco.parati@unimib.it; 2Department of Internal Medicine, IRCCS Sacro Cuore-Don Calabria Hospital, 37024 Negrar, Italy; filippo.valbusa77@gmail.com; 3CHRU-Nancy, Pôle “Maladies du Vieillissement, Gérontologie et Soins Palliatifs”, Université de Lorraine, 54000 Nancy, France; a.kearney-schwartz@chru-nancy.fr (A.K.-S.); a.benetos@chru-nancy.fr (A.B.); 4INSERM, DCAC u1116, Université de Lorraine, 54000 Nancy, France; carlos.labat@inserm.fr; 5Medicina Clinica, Azienda Sanitaria Universitaria Giuliano Isontina, 34148 Trieste, Italy; andr.grillo@gmail.com; 6Department of Medicine and Surgery, University of Milano-Bicocca, 20100 Milan, Italy

**Keywords:** aorta, arterial distensibility, arterial stiffness, augmentation index, blood pressure amplification, cardiovascular prevention, elastic modulus, pulse wave analysis, pulse wave velocity

## Abstract

Background: The stiffening of large elastic arteries is currently estimated in research and clinical practice by propagative and non-propagative models, as well as parameters derived from aortic pulse waveform analysis. Methods: Common carotid compliance and distensibility were measured by simultaneously recording the diameter and pressure changes during the cardiac cycle. The aortic and upper arm arterial distensibility was estimated by measuring carotid–femoral and carotid–radial pulse wave velocity (PWV), respectively. The augmentation index and blood pressure amplification were derived from the analysis of central pulse waveforms, recorded by applanation tonometry directly from the common carotid artery. Results: 75 volunteers were enrolled in this study (50 females, average age 53.5 years). A significant inverse correlation was found between carotid distensibility and carotid–femoral PWV (r = −0.75; *p* < 0.001), augmentation index (r = −0.63; *p* < 0.001) and central pulse pressure (r = −0.59; *p* < 0.001). A strong correlation was found also between the total slope of the diameter/pressure rate carotid curves and aortic distensibility, quantified from the inverse of the square of carotid–femoral PWV (r = 0.67). No correlation was found between carotid distensibility and carotid–radial PWV. Conclusions: This study showed a close correlation between carotid–femoral PWV, evaluating aortic stiffness by using the propagative method, and local carotid cross-sectional distensibility.

## 1. Introduction

Several studies have shown that the assessment of arterial function and structure provides prognostic information incremental to conventional cardiovascular risk stratification [1,2,3,4]. Arterial stiffness is strongly related to cardiovascular risk factors and cardiovascular morbidity and mortality [5], particularly in individuals with end-stage renal disease [6,7,8,9], in hypertensive patients [1,10,11], in diabetic patients [12], in very old individuals [13,14], and in coronary patients [2,3,15,16,17].

Different models were proposed to estimate the mechanical properties of the large arteries [18]: (i) propagative models, (ii) non-propagative models, and (iii) central pulse wave analysis.

In propagative models, arterial distensibility is defined from the propagation velocity of the pulse wave. The pulse wave runs through the arterial vessels at a speed that depends on the elasticity of the wall itself: the less elastic the wall, the greater the propagation velocity. In non-propagative models, the cross-sectional mechanical properties of the arteries can be assessed non-invasively on the basis of the volume/pressure relationship of an arterial segment [19,20]. Vascular distensibility is defined by the change in diameter in relation to the blood pressure (BP) change. It is possible to define the degree of vascular distensibility by simultaneously measuring the variation in BP and vascular diameter.

Parameters derived from the analysis of the pulse pressure (PP) waveform have been suggested as markers of arterial stiffness, such as augmentation index (AIx) [21] and BP amplification phenomenon [22]. AIx estimates the increase in systolic BP caused by an early return of reflected waves to the aorta and is defined by the difference between the second and first systolic peaks, expressed as a percentage of the central PP [22,23,24]. BP amplification is calculated from the increase of PP from the central aorta toward the periphery and is mainly attributed to systolic BP increase [22,23,25,26,27].

The purpose of the present study was to evaluate the relationship between variables derived from the above models: pulse wave analysis (AIx, PP amplification), propagative models (carotid–femoral and carotid–radial pulse wave velocity), and non-propagative models (carotid cross-sectional compliance and distensibility). Our study also aimed at determining, for each of them, the specific influence of age, sex, anthropometric parameters, and BP levels.

## 2. Materials and Methods

Participants in this study were volunteers (age range between 20 and 90 years) recruited among the medical and paramedical staff, day hospital patients or outpatients from the Geriatric Department of the University Hospital of Nancy (France), with an equal distribution in three age subgroups (20–45, 46–70, and 71–90 years). The absence of major systemic diseases was confirmed by physical and laboratory routine examinations. All the individuals recruited were given a clear explanation of the aims of the trial, and all were asked to give their consent to the study procedures. The protocol of this study was approved in Nancy, France, by the “Comité de Protection des Personnes” of Nancy CPP Est III 15 December 2006. After recording the weight and height of all recruited individuals, a rest period in supine position for 15’ for acclimatization was scheduled before data collection. Body mass index (BMI) was calculated as weight divided by height squared (kg/m^2^). A validated oscillometric sphygmomanometer Omron 705IT (Omron Co., Kyoto, Japan) [28] was used for brachial BP measurement. After BP measurement, a Duplex sonography (Esaote) was performed on both carotid arteries to exclude vascular stenosis. The presence of atherosclerotic plaques, defined as a focal intima-media thickness >1.5 mm, was an additional study exclusion criterion.

### 2.1. Carotid Distensibility and Compliance

Common carotid compliance and distensibility were measured by simultaneously recording the carotid diameter and pressure change during the cardiac cycle [29]. The diameter variation curve was recorded by the Wall Track System (Pie Medical, Maastricht, The Netherlands) [30]. This system measures the variation of the carotid diameter during the cardiac cycle, using a radio frequency analysis implemented in the Esaote ultrasound system (Genoa, Italy) [31,32]. For this study, the Wall Track System was programmed to simultaneously record the vascular diameter variation curve and the electrocardiogram (ECG) tracing for a duration of 4 s, with a 200 Hz sampling rate (one signal acquired every 5 ms). A validated PulsePen (DiaTecne s.r.l., San Donato Milanese, Italy) [33,34,35] transcutaneous arterial tonometer was used to record the carotid BP wave. The PulsePen simultaneously records the pressure curve and the ECG tracing for a period of 10 cardiac cycles, with a 1 kHz sampling rate.

The synchronization of the two waves (diameter and pressure) is allowed by superimposing the ECG readings recorded at the same time as the two curves via a specifically software DiaPres (DiaTecne s.r.l., San Donato Milanese, Italy). This method was previously described in detail [29]. The change in diameter/change in pressure rate and the slope of the curve concerning this rate were estimated from the entire cardiac cycle (Figure 1), from the proto-mesosystolic phase (a–b phase in Figure 1), and from the diastolic phase (c–a’ phase in Figure 1).

Common carotid arterial cross-sectional compliance was calculated according to the following formula [19,36,37]:$$Compliance=\frac{D_{max}-D_{min}}{PP}$$
where *D*_max_ and *D*_min_ are the maximal and minimal calculated carotid diameters obtained during a cardiac cycle, and *PP* is the carotid *PP* measured by arterial tonometer.

Arterial cross-sectional distensibility is the compliance value normalized for the carotid cross-sectional diameter:$$Distensibility=\frac{D_{max}-D_{min}}{PP\times D_{min}}$$

Compliance and distensibility define, respectively, absolute and relative diameter change for every 1 mmHg increase in BP.

Common carotid elastic modulus was determined by the following formula:$$Elastic\;Modulus\;=\frac{PP}{S_{AD}}$$
where *S_AD_* is the strain of the common carotid diameter, defined as follows:$$S_{AD}=\frac{\left(D_{\max}-D_{\min}\right)}{D_{\min}}$$

Carotid diameter measurements were obtained by assuming circular geometry of the common carotid artery. Elastic modulus is the pressure change required for theoretical 100% stretch from resting diameter. It is, thus, the inverse of distensibility.

The stiffness index was calculated according to the following formula:$$Stiffness\;Index\;=\frac{ln(SBP/DBP)}{S_{AD}}$$
where *SBP* and *DBP* are carotid systolic and diastolic BP.

Two recordings were performed in quick succession. From the analysis of these diameter and pressure curves, we calculated the mean value of the arterial wall properties’ parameters required by our study protocol.

### 2.2. Pulse Wave Velocity (PWV)

The carotid, femoral, and radial pressure waves were recorded with the PulsePen tonometer to measure the carotid–femoral pulse wave velocity (cf-PWV) and the carotid–radial pulse wave velocity (cr-PWV). PWV, measured in m/s, is defined by the relationship between the distance travelled by the pressure wave and the delay in registering the distal wave (femoral artery for cf-PWV, and radial artery for cr-PWV) compared with the carotid wave [38]. The distance travelled was assessed by subtracting carotid to suprasternal notch distance from suprasternal notch to peripheral (femoral or radial) distance. We used the mathematical model proposed by Bramwell and Hill [39] to relate the arterial wall distensibility with the inverse of the square of pulse wave velocity.

### 2.3. Pulse Wave Analysis and Central Blood Pressure Measurement

Central BP values and central pulse pressure waveforms were recorded directly from the common carotid artery, using a validated applanation tonometer [40,41,42] (PulsePen device). This device was described in detail elsewhere [33,34]. As previously demonstrated, the pressure waves recorded non-invasively by the PulsePen tonometer at the site of the common carotid artery are almost equal to the pressure waveforms obtained invasively by means of an intra-arterial catheter in the same arterial segment [33]. Moreover, several studies have demonstrated that central BP values and pulse wave parameters recorded in the common carotid artery are reliable surrogates of the corresponding parameters recorded in the aorta by invasive methods [19,23,33]. Central BP values were obtained from carotid BP curve analysis and brachial BP measurements, using an appropriate validated algorithm [19,33]. The amplification phenomenon (from aorta to brachial artery) was expressed as (i) PP amplification rate, i.e., the percentage of increase of PP in the brachial artery (PP_B_) relative to central PP (PP_C_), according to the formula PPA = 100·(PP_B_ − PP_C_)/PP_C_; and (ii) systolic BP amplification, i.e., the difference between brachial systolic BP and aortic systolic BP, expressed in mmHg. We defined AIx as the difference between the second and first systolic peaks of carotid pressure waveform and expressed as a percentage of central PP.

### 2.4. Statistical Analysis

All statistical analyses were performed with NCSS version 9 Statistical Software (NCSS, LLC, Kaysville, UT, USA). Values are presented as means ± standard deviation. The relationship between hemodynamic parameters was tested with Spearman’s correlation coefficient in univariate analysis. A simple regression test was performed for the analysis of bivariate linear correlations. A *p* < 0.05 was considered as the level of statistical significance. As a measure of BP for multiple regression analysis, we used only mean arterial pressure, because systolic BP and PP are influenced by arterial stiffness. Multivariate stepwise regression analyses were performed, including age, sex, weight, height, mean arterial pressure, and heart rate as independent variables, while arterial-stiffness-related variables were taken as dependent variables. In multivariate models, the *p*-to-enter was set at 0.10, and the *p*-to-stay to 0.05.

### 2.5. Preliminary Study: Standardization of the Method for Measuring Vascular Distensibility

We performed a preliminary study to overcome the difficulties in analyzing the diameter and pressure curves simultaneously in the same carotid artery before beginning a comparative study between the various methods. We explored whether the simultaneous acquisition of the diameter and pressure curves on the same carotid artery could be adequately replaced by the simultaneous acquisition of the same two curves on the two carotid arteries by deriving the pressure curve from the right carotid artery and the diameter curve from the left carotid artery. In fact, using both carotid arteries would make the test considerably simpler to perform.

Thus, in the first 10 healthy volunteers included in our study (3 males and 7 females, mean age of 30.0 ± 4.7 years), the diameter and pressure curves were first recorded simultaneously on the same carotid artery (right carotid artery), placing the tonometer in a distal position in relation to the ultrasound probe. Subsequently, the systolic–diastolic variations in diameter and pressure were simultaneously recorded on the two common carotid arteries, 2–3 cm proximal to the carotid bifurcation, by placing the ultrasound probe on the left common carotid, and the tonometer on the right common carotid. The results obtained from a single carotid artery and those obtained from both left and right carotid arteries were then compared. A data analysis was performed according to the recommendations of Bland and Altman [43]. The relative (positive or negative) differences between each pair of measurements were plotted against their mean to evaluate the relationship between the mean value of the considered parameter and the difference in the estimates provided by these two approaches. The level of agreement between the two series of measurements was estimated by their mean difference and the standard deviation of these differences [43]. The coefficient of variation was defined as the standard deviation of the differences between the methods divided by the mean of the absolute difference values.

The study highlighted the high reliability of the data obtained by positioning the two probes on the two carotid arteries. The mean difference between the values obtained by using the two approaches was always very close to zero, and the differences between the two series of data that appeared were homogeneously distributed across each parameter value distribution and were, in any case, always less than 10% of the mean values. As an example, for a mean value of arterial distensibility of 3.77 ± 0.71, the mean of the differences ± twice the standard deviation was 0.06 ± 0.33. The coefficient of variation values were lower than 5% for the parameters of vascular distensibility (coefficient of variation 4.5%) and vascular compliance (coefficient of variation 4.4%). Based on these initial results, we then decided to use recordings performed on both carotid arteries in our study. This choice was supported also by the fact that the test performed on both carotid arteries is considerably simpler to carry out, and the signal is easier to obtain, and its reliability is supported by another study [44]. Moreover, using both arteries avoids the possibility of technical interferences with the measurement, due to, for example, the slight compression of the carotid artery by the tonometer, which might affect ultrasound diameter assessment; or the interference between the gel used for the ultrasound recordings and the receiving end of the tonometric probe.

## 3. Results

A total of 75 volunteers were enrolled in the study (66% females, average age 53.5 years). Seven individuals (9.3%) were smokers, and none had a history of alcohol consumption, although three reported occasional drinking. Table 1 summarizes the clinical, hemodynamic, and anthropometric parameters.

The BP parameters, PP amplification, PWV and carotid cross-sectional distensibility appeared to be significantly correlated with age. The relationships of cf-PWV and cr-PWV with age were quite different. In the young age group, the cf-PWV values were lower than the cr-PWV values (in first tertile, cf-PWV was 6.350 ± 0.99 m/s, and cr-PWV was 7.76 ± 1.75 m/s). With aging, the cr-PWV increased much less than the cf-PWV (in third tertile, cf-PWV was 15.33 ± 4.13 m/s, and cr-PWV was 8.72 ± 1.24 m/s). Figure 2 clearly shows that the cf-PWV values in third tertile were more than two times greater than in the first tertile and increased quickly in the elderly. Thus, the relationship between age and PWV is more appropriately expressed by a quadratic non-linear model than by the conventional linear model approach.

Figure 3 shows the changes in carotid artery compliance curves with age. In this figure, only the first and third tertiles of age are shown: all curves of the youngest individuals are positioned in the upper and left part of the figure (small BP changes determine large changes in vascular diameter); on the contrary, curves related to the oldest people are positioned in the lower and right part of the figure (large pressure changes determine small changes in diameter).

The global (systolic and diastolic) slopes of the diameter/pressure rate curves related to the different tertiles of age are shown in Figure 4. All slope values in the first tertile are higher than the mean value of the second, and all values of the third tertile are lower than mean values of the second tertile.

Table 2 shows the results of the bivariate analysis between PP amplification, AIx, cf-PWV, cr-PWV, and carotid artery distensibility and age, sex, anthropometric parameters, BP values, and heart rate. Height was directly correlated with carotid distensibility (*p* < 0.02) and inversely correlated with AIx (*p* < 0.001); BMI showed a significant inverse correlation with carotid distensibility (*p* < 0.001), cf-PWV (*p* < 0.001) and cr-PWV (*p* < 0.04). All the studied parameters showed a significant strong correlation with central and peripheral systolic BP and PP (*p* < 0.001), with the exception of cr-PWV, which instead appeared to be correlated with diastolic and mean BP. Heart rate was significantly correlated with PP amplification (*p* < 0.004) and inversely correlated with AIx (*p* < 0.03).

Table 3 shows the results of bivariate analysis between local carotid artery hemodynamic parameters and measurements derived from the central pulse wave analysis and by propagative models (cf-PWV). A strong inverse correlation was found between carotid distensibility and cf-PWV (r = −0.75; *p* < 0.001), AIx (r = −0.63; *p* < 0.001), and central PP (r = −0.59; *p* < 0.001). A significant, but weaker, correlation was found also between carotid distensibility and PP amplification (r = 0.34; *p* < 0.003) and cr-PWV (inverse correlation, r = −0.39; *p* < 0.001). Finally, an inverse relationship was found between carotid distensibility and amplification of systolic BP, expressed as the difference between brachial and carotid systolic BP (r = −0.24; *p* < 0.04).

The relationship between slope of the diameter/pressure rate curves across the entire cardiac cycle and aortic distensibility, quantified as the inverse of (cf-PWV)^2^, is shown in Figure 5. AIx was strongly correlated with cf-PWV (r = 0.53; *p* < 0.001) and weakly related to cr-PWV (r = 0.29; *p* < 0.02).

Table 4 shows the results of the multivariate stepwise regression analysis with cf-PWV, cr-PWV, AIx, PP amplification, amplification of systolic BP, and carotid arterial distensibility as dependent variables, and age, sex, heart rate, mean BP, height, and weight as independent variables. None of the latter parameters was a determinant of cr-PWV. Age was the main factor determining cf-PWV, Aix, and carotid distensibility. Heart rate was the main variable determining PP amplification and was a significant independent variable also for AIx and carotid distensibility. Sex was a significant determinant only for AIx. Mean BP was the main variable positively associated with systolic BP amplification. Mean BP was also inversely associated with carotid distensibility and directly associated with AIx.

## 4. Discussion

A number of techniques are currently used to estimate arterial stiffness in a clinical setting, with limited attention to the differences among the methods being used. In such a context, our study offers clear pathophysiological data, based on the simultaneous use of different approaches to the estimate of arterial distensibility, that might help clinicians in correctly interpreting the results of commercially available devices. The close inverse correlation found in our study between cf-PWV, estimated through a “propagative method”, and carotid distensibility, estimated through a cross-sectional method, confirms the close link between these two methods. The estimates of aortic and carotid distensibility were also correlated with the major factors which are known to determine alterations in the viscoelastic properties of the large arteries, particularly age and high BP levels.

In spite of these correlations, however, these two approaches offer different perspective on the assessment of mechanical vascular wall properties across the arterial tree.

Overall, cf-PWV accurately reflects the speed of BP wave propagation across the arterial tree from the heart to the periphery [34]. Depending on the PWV itself and on the distance covered, the reflected wave generated at the periphery will overlap with the forward BP wave at different times during the cardiac cycle. In the presence of a low stiffness-low PWV state, the reflected waves are overlapped with the forward waves during the early systolic phase of the cardiac cycle in the peripheral arteries, while in the central arteries, this overlapping occurs during the late systolic phase. Therefore, reflected waves will not contribute to increasing central systolic and PP, while they will amplify peripheral systolic BP and PP. This mechanism explains the PP amplification phenomenon, i.e., why peripheral (brachial) PP is higher than central (aortic or carotid) PP in the presence of elastic arteries.

Indeed, cf-PWV is currently considered as the “gold-standard” measurement of large arteries stiffness [18,45], because, according to the mathematical model proposed by Bramwell and Hill [39], arterial wall distensibility is related to the reverse of the pulse wave propagation velocity squared.

We compared the aortic distensibility estimated by cf-PWV with the assessment of carotid distensibility provided by a carotid cross-sectional approach (wall track system). Although carotid cross-sectional distensibility was obtained only from a specific and well-defined segment of the artery, our study points out that, in the absence of evident atheromatous disease, the hemodynamic conditions and the viscoelastic properties of the common carotid artery are similar to those of the aorta, as shown by the inverse correlation we found between cf-PWV one side and carotid distensibility and AIx on the other side. The mechanical properties of a blood vessel are not linear, i.e., they depend on the pressure distending them, which varies continuously The simultaneous measurement of diameter and pressure variation curves for the definition of carotid cross-sectional distensibility obtained in our study is, thus, an important test, complementary to the tests that are commonly used to study the mechanical properties of the large arteries and the degree of vascular ageing, such as PWV and aortic pulse wave analysis.

In spite of the complementary nature of aortic pulse wave analysis and propagative and non-propagative models, the factors affecting these parameters are different. Overall, cf-PWV is determined by age; AIx by age, gender, heart rate, and BP; PP amplification by heart rate and age; and carotid cross-sectional distensibility is influenced by age, BP, weight, and heart rate.

A further element that has emerged from our study is the inadequacy of the study of the axillo–brachial–radial axis as a window for the general evaluation of the viscoelastic properties of the large arteries. In fact, we found only a weak relationship between cr-PWV on one side, and age, BP, carotid distensibility and compliance and cf-PWV on the other side. The results of our study, therefore, offer additional and complementary evidence supporting and expanding previous observations regarding the fact that the arterial tree is not homogeneous and the various arterial districts can be differently affected either by the aging process, by hypertension or by a combination of both factors [46,47]. Formerly, also, van der Heijden-Spek et al. [48] showed that, after the adjustment for the confounding factor, no relation exists between the age and distensibility of the brachial artery. Moreover, it has been shown that, for the same mean transmural pressure, normotensive and hypertensive patients had the same PWV in the forearm and, therefore, the same distensibility [49]. Evidence is also available that radial artery compliance paradoxically increases in hypertensive patients, compared to normotensive individuals, when assessed at the same BP level [50,51]. This can be related to the specific anatomical characteristics of the arteries of the upper limb, which are prevalently muscular (whereas the aorta has a mixed, prevalently elastic structure, especially in its proximal portion) and not subjected to vascular ageing.

Our study has a few limitations affecting the generalizability of our results. Some caution should be used in the interpretation of the data, due to limited sample of the subgroups, which makes a complete adjustment for confounders in the subgroup analysis impossible. A further limitation consists of the selection of individuals considered in the present study, which is lacking patients with overt cardiovascular disease, to whom the evaluation of arterial stiffness is usually addressed. Larger studies in cardiovascular patients are needed to confirm our data.

## 5. Conclusions

In this study, we expand the available evidence on the close association between aortic propagative models detecting arterial stiffness (cf-PWV) and local carotid cross-sectional models (based on simultaneous measurement of diameter and pressure variation curves, in order to measure distensibility and compliance). Instead, cr-PWV clearly appeared in our study to be inadequate for the general evaluation of the mechanical properties of large arteries. These data strongly support the complementary nature of some, but not all, of the currently available methods to estimate mechanical arterial wall properties in a clinical setting.

## Figures and Tables

**Figure 1 jcm-11-02225-f001:**
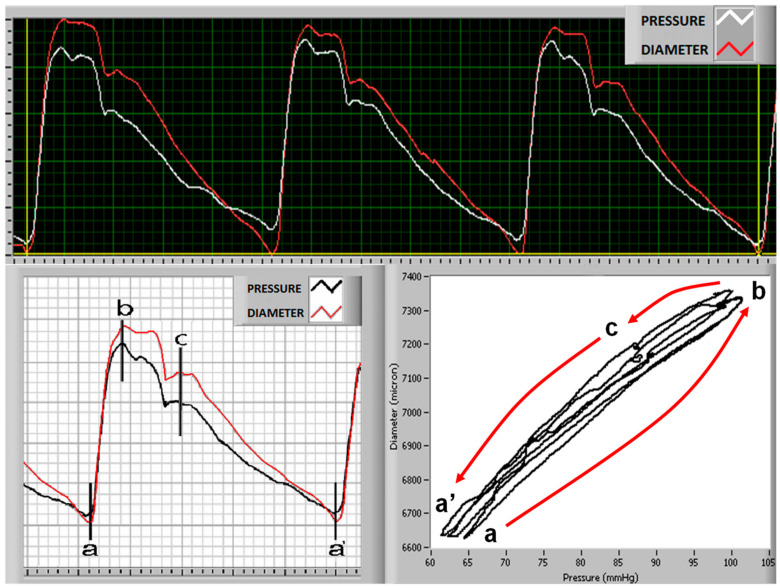
Upper panel: original tracings of carotid diameter (red line) and simultaneously acquired blood pressure (white line) wave. Lower panel: on the left arterial pressure wave (black line) and curves of cross-sectional diameter changes (red line); a–b interval indicates the proto-mesosystolic phase, and c–a’ is the late diastole phase (lower panel). Cross-sectional diameter/pressure curves are shown in the right lower panel.

**Figure 2 jcm-11-02225-f002:**
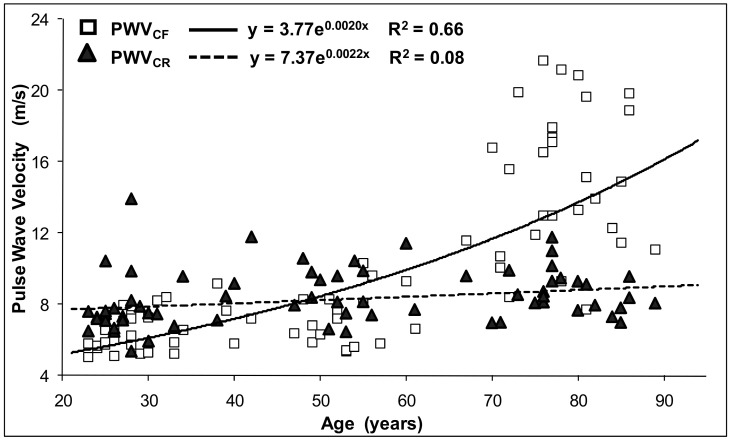
Relationship between age and carotid–femoral pulse wave velocity PWV_CF_ (continuous line, open squares) and carotid–radial pulse wave velocity PWV_CR_ (dotted line, closed triangles).

**Figure 3 jcm-11-02225-f003:**
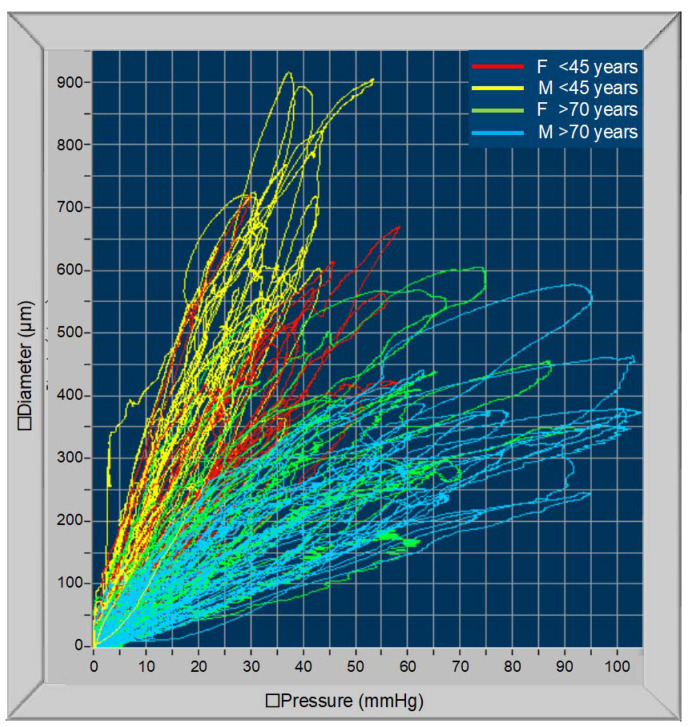
Carotid cross-section pressure curves in individuals aged 20–45 years (red lines, females; yellow lines, males) and over 70 years old (green lines, females; blue lines, males).

**Figure 4 jcm-11-02225-f004:**
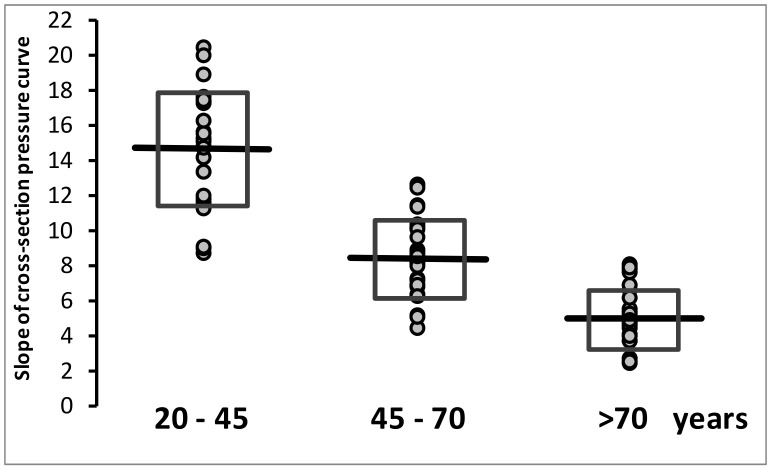
Slope of the cross-section diameter–pressure curves are separately shown for the different age tertiles. Mean values and standard deviation (open rectangles) are shown.

**Figure 5 jcm-11-02225-f005:**
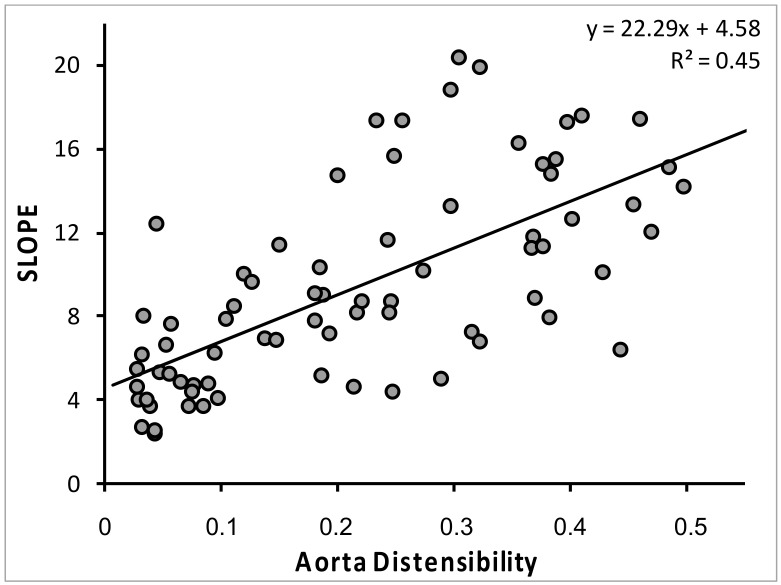
Univariate linear relationship between the slope of the cross-section carotid diameter-pressure curves and aorta distensibility [1/(carotid–femoral PWV)^2^].

**Table 1 jcm-11-02225-t001:** Clinical, anthropometric, and hemodynamic parameters.

Parameter	Pooled	Age Groups (years)			Trend
		20–45	46–70	>70	*p*
Subjects	75	25	25	25	
Gender, M/F	25/50	9/16	8/17	8/17	
BMI, kg/m^2^	24.4 ± 3.4	23.3 ± 3.6	24.9 ± 3.4	24.9 ± 3.3	n.s.
Height, cm	166.9 ± 9.3	170.3 ± 8.8	166.9 ± 9.0	163.0 ± 9.0	0.03
Weight, kg	68.0 ± 12.2	67.6 ± 10.4	69.7 ± 13.5	66.6 ± 12.8	n.s.
Brachial Systolic BP, mmHg	126.4 ± 20.0	112.6 ± 11.2	124.2 ± 13.5	142.2 ± 21.4	<0.0001
Central Systolic BP, mmHg	114.3 ± 18.4	101.0 ± 11.2	113.0 ± 12.1	128.8 ± 19.2	<0.0001
Brachial PP, mmHg	55.7 ± 17.0	47.1±7.9	49.6 ± 11.5	70.4 ± 18.7	<0.0001
Central PP, mmHg	43.6 ± 15.5	35.4 ± 8.0	38.3 ± 10.6	57.0 ± 16.6	<0.0001
Mean BP, mmHg	89.2 ± 11.7	81.3 ± 7.1	91.2 ± 8.6	95.3 ± 13.8	<0.0001
Diastolic BP, mmHg	70.± 10.1	65.6 ± 5.8	74.6 ± 8.0	71.8 ± 13.2	0.005
LVET, ms	296.1 ± 28.9	294.8 ± 22.0	303.4 ± 24.7	290.1 ± 37.3	n.s.
Diastolic Time, ms	592.0 ± 116.6	594.0 ± 101.0	575.2 ± 131.2	606.9 ± 118.2	n.s.
Heart Rate, bpm	69.1 ± 10.3	68.6 ± 8.7	70.1 ± 11.9	68.4 ± 10.4	n.s.
Amplification, mmHg	12.1 ± 4.3	11.6 ± 2.6	11.2 ± 4.7	13.4 ± 5.0	n.s.
PP Amplification, %	30.1 ± 11.5	34.6 ± 10.6	31.1 ± 12.9	24.8 ± 8.8	0.008
AIx, %	12.8 ± 20.7	−8.5 ± 11.7	20.1 ± 16.7	26.7 ± 13.1	<0.0001
cf-PWV, m/s	9.90 ± 4.81	6.35 ± 0.99	8.03 ± 2.38	15.33 ± 4.13	<0.0001
cr-PWV, m/s	8.41 ± 1.58	7.76 ± 1.75	8.82 ± 1.51	8.72 ± 1.24	0.04
Carotid cross-sectional					
Compliance, μm/mmHg	9.79 ± 4.89	15.28 ± 3.36	8.57 ± 2.22	5.54 ± 2.41	<0.0001
Distensibility, mmHg^−1^	1.50 ± 0.86	2.49 ± 0.55	1.27 ± 0.37	0.74 ± 0.38	<0.0001
Elastic Modulus, mmHg	977 ± 663	427 ± 127	864 ± 276	1641 ± 691	<0.0001
Stiffness Index	4.26 ± 2.49	2.17 ± 0.61	3.77 ± 1.07	6.84 ± 2.40	<0.0001
Carotid Global Slope	9.4 ± 4.7	14.7 ± 3.3	8.2 ± 2.1	5.3 ± 2.2	<0.0001
Carotid Systolic Slope	9.7 ± 4.9	15.1 ± 3.6	8.7 ± 2.3	5.3 ± 1.9	<0.0001
Carotid Diastolic Slope	11.9 ± 5.9	18.4 ± 4.1	10.4 ± 3.3	6.9 ± 2.9	<0.0001

Data are shown for the whole cohort (first column), and by dividing the cohort in tertiles of age. Last column shows the statistical significance of the parameter variation trend as a function of age. Data are expressed as means ± standard deviations. AIx, augmentation index; BMI, body mass index; BP, blood pressure; cf-PWV, carotid–femoral pulse wave velocity; cr-PWV, carotid–radial pulse wave velocity; LVET, left ventricular ejection time; M/F, males/females; n.s., not significant; PP, pulse pressure.

**Table 2 jcm-11-02225-t002:** Bivariate analysis (Spearman’s correlation coefficient) between the main parameters for estimating arterial stiffness and clinical and anthropometric parameters.

Parameter	PPA		AIx		cf-PWV		CCS Distensibility	
	r	*p*	r	*p*	r	*p*	r	*p*
Sex	0.05	n.s.	0.33	0.005	−0.13	n.s.	0.02	n.s.
Age	−0.47	<0.001	0.76	<0.001	0.79	<0.001	−0.84	<0.001
BMI	−0.18	n.s.	0.15	n.s.	0.38	0.001	−0.40	<0.001
Height	0.19	n.s.	−0.53	<0.001	−0.14	n.s.	0.28	0.02
Weight	−0.03	n.s.	−0.26	0.03	0.17	n.s.	−0.09	n.s.
bSBP	−0.35	0.002	0.49	<0.001	0.59	<0.001	−0.67	<0.001
cSBP	−0.47	<0.001	0.52	<0.001	0.58	<0.001	−0.66	<0.001
bPP	−0.48	<0.001	0.41	<0.001	0.54	<0.001	−0.59	<0.001
cPP	−0.67	<0.001	0.46	<0.001	0.53	<0.001	−0.59	<0.001
MAP	−0.13	n.s.	0.40	<0.001	0.50	<0.001	−0.57	<0.001
DBP	0.10	n.s.	0.24	0.04	0.29	0.01	−0.35	0.002
LVET	−0.25	0.03	0.31	0.007	−0.24	0.04	0.16	n.s.
DT	−0.35	0.002	0.22	0.05	0.02	n.s.	0.08	n.s.
HR	0.33	0.004	−0.26	0.03	0.05	n.s.	−0.11	n.s.

AIx, augmentation index; BMI, body mass index; bPP, brachial pulse pressure; bSBP, brachial systolic blood pressure; CCS, carotid cross-sectional; cf-PWV, carotid–femoral pulse wave velocity; cPP, carotid pulse pressure; cSBP, carotid systolic blood pressure; DBP, diastolic blood pressure; DT, diastolic time; HR, heart rate; LVET, left ventricular ejection time; MAP, mean arterial pressure; n.s., not significant; PPA, pulse pressure amplification.

**Table 3 jcm-11-02225-t003:** Bivariate analysis (Spearman’s correlation coefficient) between carotid cross-sectional distensibility measurements and main hemodynamic parameters, derived from central pulse wave analysis, and carotid–femoral PWV.

Carotid Cross-Sectional Measurements	Peripheral PP		Central PP		PP Amplification		AIx		cf-PWV	
	r	*p*	r	*p*	r	*p*	r	*p*	r	*p*
Distensibility	−0.67	<0.001	−0.59	<0.001	0.34	0.003	−0.63	<0.001	−0.75	<0.001
Compliance	−0.65	<0.001	−0.59	<0.001	0.34	0.003	−0.65	<0.001	−0.72	<0.001
Elastic Modulus	0.67	<0.001	0.59	<0.001	−0.34	0.003	0.63	<0.001	0.75	<0.001
Stiffness Index	0.54	<0.001	0.51	<0.001	−0.33	0.004	0.61	<0.001	0.74	<0.001
Total slope	−0.65	<0.001	−0.59	<0.001	0.36	0.002	−0.65	<0.001	−0.71	<0.001
Systolic slope	−0.66	<0.001	−0.59	<0.001	0.32	0.005	−0.62	<0.001	−0.75	<0.001
Diastolic slope	−0.61	<0.001	−0.53	<0.001	0.25	0.03	−0.61	<0.001	−0.68	<0.001

AIx, augmentation index; cf-PWV, carotid–femoral pulse wave velocity; PP, pulse pressure.

**Table 4 jcm-11-02225-t004:** Results of stepwise regression analysis with the main parameters for estimating arterial stiffness as dependent variables and anthropometric and clinical parameters as independent variables.

Dependent Variable	r^2^	Independent Variable	Regression Coefficient	SE	Β	*p*	r^2^ Change(%)
SBP Amplification	0.29	MAP	0.134	0.045	0.37	0.004	15.3
mmHg		HR	0.120	0.046	0.29	0.01	7.4
PP Amplification	0.31	HR	0.365	0.118	0.33	0.003	10.2
		Age	−0.154	0.067	−0.30	0.02	14.9
AIx	0.72	Age	0.534	0.077	0.59	<0.001	52.8
		HR	−0.530	0.136	−0.27	<0.001	6.6
		Sex	14.435	4.235	0.35	0.001	10.1
		MAP	0.381	0.133	0.22	0.006	1.6
cf-PWV	0.66	Age	0.169	0.020	0.79	<0.001	61.4
CCS Distensibility	0.79	Age	−0.030	0.003	−0.76	<0.001	70.8
		Weight	−0.017	0.006	−0.24	0.003	3.9
		MAP	−0.012	0.005	−0.16	0.02	3.0
		HR	−0.011	0.005	−0.13	0.04	1.4

Regression coefficient quantifies the slope of the regression line, and β provides a measure of the relative strength of the association independent of the measurement units. AIx, augmentation index; CCS, carotid cross-sectional; cf-PWV, carotid–femoral pulse wave velocity; HR, heart rate; MAP, mean arterial pressure; PP, pulse pressure; SBP, systolic blood pressure; Sex is 1 = males and 2 = females.

## Data Availability

The data presented in this study are available upon request from the corresponding author. The data are not publicly available, due to privacy concerns.

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
