# Peer review of "Non-Invasive Assessment of Arterial Stiffness: Pulse Wave Velocity, Pulse Wave Analysis and Carotid Cross-Sectional Distensibility: Comparison between Methods"

_jcm, 2022, doi:10.3390/jcm11082225_

Round 1
Reviewer 1 Report
This analysis is interesting.
I would include a limitations section in the Discussion. This should include, but not limited to, caution when interpreting the data due to limited sample in the subgroups, as well as the statistical methods that cannot fully adjust for a small sample size.
Author Response
As kindIy suggested, in our new version of our paper, we have included a limitations section in the Discussion. We have revised the entire manuscript and significantly improved the English language.

Reviewer 2 Report
The study states that the arterial system is not homogeneous and different arterial segments may be differently affected by the aging process, high blood pressure or other factors. Thus, the relationship between age and PWV is more appropriately expressed by a nonlinear quadratic model than by the conventional linear model approach. Carotid artery distensibility can be influenced by age, blood pressure values, heart rate. BMI showed a reverse significant correlation with carotid distensibility.
Bivariate analysis between local hemodynamic parameters of the carotid artery and the measurements derived from the analysis of central pulse waves and the propagation patterns (cf-PWV) showed a reverse correlation between carotid distensibility and cf-PWV, AIx and central PP. Sex was a significant determinant for AIx alone.
The practical utility of the study lies in highlighting the correlation between carotid-femoral PWV that evaluates the propagating aortic stiffness and the distensibility of the carotid cross section. A correlation is also made between aortic and carotid distensibility.
They also concluded that the study of axillo-brachial-radial axis to be considered as a window for the general evaluation of the viscoelastic properties of the large arteries. Moreover, it has been shown that, for the same mean transmural pressure, normotensive and hypertensive patients had the same PWV in the forearm and therefore the same distensibility. The article is well written and documented.
Author Response
We thank the reviewer for the nice and detailed analysis of our paper. As kindIy suggested, we have revised the entire manuscript and significantly improved the English language.
